# PI3K Signaling in Mechanisms and Treatments of Pulmonary Fibrosis Following Sepsis and Acute Lung Injury

**DOI:** 10.3390/biomedicines10040756

**Published:** 2022-03-23

**Authors:** Jean Piero Margaria, Lucia Moretta, Jose Carlos Alves-Filho, Emilio Hirsch

**Affiliations:** 1Department of Molecular Biotechnology and Health Sciences, Molecular Biotechnology Center, University of Torino, Via Nizza 52, 10126 Torino, Italy; jeanpiero.margaria@unito.it (J.P.M.); lucia.moretta@edu.unito.it (L.M.); 2Department of Pharmacology, Ribeirão Preto Medical School, University of São Paulo, Avenida Bandeirantes 3900, Ribeirao Preto 14049-900, Brazil; jcafilho@usp.br

**Keywords:** pulmonary fibrosis, PI3K, sepsis, ARDS, inhibitor

## Abstract

Pulmonary fibrosis is a pathological fibrotic process affecting the lungs of five million people worldwide. The incidence rate will increase even more in the next years due to the long-COVID-19 syndrome, but a resolving treatment is not available yet and usually prognosis is poor. The emerging role of the phosphatidylinositol 3-kinase (PI3K)/AKT signaling in fibrotic processes has inspired the testing of drugs targeting the PI3K/Akt pathway that are currently under clinical evaluation. This review highlights the progress in understanding the role of PI3K/Akt in the development of lung fibrosis and its causative pathological context, including sepsis as well as acute lung injury (ALI) and its consequent acute respiratory distress syndrome (ARDS). We further summarize current knowledge about PI3K inhibitors for pulmonary fibrosis treatment, including drugs under development as well as in clinical trials. We finally discuss how the design of inhaled compounds targeting the PI3K pathways might potentiate efficacy and improve tolerability.

## 1. Introduction

Pulmonary fibrosis includes a heterogenous group of disorders characterized by the irreversible replacement of lung parenchyma with fibrotic tissue, progressing to organ malfunction and eventually death from respiratory failure [1]. Disease progression consists in a gradual substitution of the parenchymal pulmonary tissue with a fibrotic scar [2]. The fibrotic remodeling of the lungs may originate from several disorders, all characterized by an increased collagen deposition in loco that evolves into pulmonary fibrosis and eventually into lung stiffening. One of the commonest forms of pulmonary fibrosis is idiopathic pulmonary fibrosis (IPF), a pathological condition affecting the lungs of five million people worldwide [3]. Nonetheless, pulmonary fibrosis can also result from acute lung injury (ALI) eventually triggering acute respiratory distress syndrome (ARDS), often as a complication of severe sepsis caused by bacterial and viral infection. This condition is frequently worsened by the development of an immune system crisis derived from the uncontrolled secretion of soluble signaling factors, leading to the so-called cytokine storm [4]. Although distinct factors triggering immune system activation and fibrogenesis determine pulmonary fibrosis following sepsis and/or ALI, such signals trigger their responses through common mechanisms, converging on a limited number of signaling pathways. This review will thus focus on one key effector of multiple cytokines and growth factors, the PI3K/Akt pathway.

PI3Ks are heterodimeric enzymes composed of a regulatory subunit coupled to a catalytic subunit, called p110. While p110α and p110β are ubiquitously expressed, p110δ and p110γ are exclusively found in leukocytes [5]. Multiple biological processes, such as cell survival, cell cycle progression, differentiation, senescence, and metabolism, affecting key events in the inflammatory response to damage and infection, are influenced by the state of activation of PI3Ks [6]. In pulmonary fibrosis following sepsis and/or ALI, multiple mediators elicit pathogenic lung remodeling through PI3K pathway activation at multiple stages of disease development, starting from the inflammatory reaction often initiating the process. For example, in the onset of sepsis and ARDS, an overwhelming stimulation of the immune system is triggered by soluble factors that activate PI3K signaling in immune cells. Further to this response, leukocytes are recruited to lung alveoli, where they promote resident fibroblasts proliferation and collagen deposition, two processes maintained by transforming growth factor-β (TGF-β) stimulation and depending on downstream activation of the PI3K signaling cascade [7,8,9]. For these reasons, targeting the PI3K pathway may mitigate several distinct steps leading to fibrotic progression and might represent a possible therapeutic option, providing promising treatment strategies for sepsis- and ARDS-induced pulmonary fibrosis.

## 2. PI3K Signaling in Sepsis Due to Bacterial and Viral Infections

Pulmonary fibrosis is one of the most common long-term complications of severe sepsis, a clinical syndrome of extreme host response to infection. Systemic effects of the syndrome afflict multiple organs, and, in the lungs, severe sepsis causes inflammation and injury that might eventually culminate in pulmonary fibrosis (Figure 1).

The two commonly used animal models of bacterial sepsis are the cecal ligation and puncture (CLP), and the treatment with lipopolysaccharide (LPS), a large molecule found in the outer membrane of Gram-negative bacteria [10]. Both pathological models trigger systemic activation of the innate immune system, lung inflammation and pulmonary fibrosis [11]. In a CLP model of sepsis, PI3K/Akt activation promotes infiltration of inflammatory cells in the lungs through the modulation of endothelial cell damage. Accordingly, treatment with Ly294002, a pan PI3K inhibitor, limits the severity of sepsis by reducing CD40L-mediated endothelial cell damage and, therefore, suppressing the extravasation of inflammatory cells in the lung parenchyma [10]. Persistent lung inflammation damages alveolar epithelial cells, which are in turn replaced by a fibrotic scar made of fibroblasts and collagen. The PI3K/AKT pathway plays a critical role in lung fibroblasts-mediated collagen deposition after LPS-induced sepsis in both murine and human cells. In mice, sepsis promotes collagen synthesis in lung fibroblasts through the activation of the PI3K-Akt-mTOR/PFKFB3 pathway, eventually leading to pulmonary fibrosis. In humans, the lung fibroblast MRC-5 cell line treated with LPS activates the PI3K-Akt-mTOR pathway to promote collagen synthesis. This process is reversed treating the cells with the PI3K inhibitor LY294002 [12].

The PI3K/AKT/mTOR signaling is critical not only in sepsis induced by bacterial infection but also in the host response to several viral pathogens, including coronavirus infection [13]. Accordingly, SF2523, a mammalian target of rapamycin (mTOR) PI3K-α/(BRD2/BRD4) inhibitor, has been proposed to prevent severe coronavirus disease 2019 (COVID-19) evolution [14]. In the context of the COVID-19 epidemic, some patients undergo severe deterioration of the tissues closely related to the cytokine storm, a condition unleashing massive production of mediators triggering the PI3K pathway. The high levels of chemokines and cytokines are mechanistically linked to higher morbidity and lethality of the infection [15]. Among the cytokines that are increased in the serum in patients with severe COVID-19, it is important to highlight IL-8 and sCD40L [16]. IL-8 is a pro-inflammatory cytokine produced in lung tissue upon lung injury. Interestingly, in an in vitro model of sepsis, the PI3K specific inhibitor LY294002 prevents inflammation by suppressing the production of IL-8 in lung epithelial cells A549 [17]. In addition, the increase in cytokine sCD40L, the soluble form of CD40L, is responsible for endothelial cell damage and leukocyte infiltration in the alveolar space, and the inhibition of the PI3K/AKT pathway through LY294002 suppresses its production [11]. These results suggests that COVID-19 disease severity is increased by the appearance of a cytokine storm, and the suppression of responses to inflammatory cytokines through PI3K inhibition may prevent the detrimental consequences for the lungs of this exaggerated endogenous response.

## 3. PI3K Signaling in ARDS and Associated Lung Fibrogenesis

Sepsis is the most common cause of ARDS, and accounts for half of all death cases [18]. In particular, sepsis causes ALI and ARDS in 6–42% of the cases [19]. In the Berlin definition (JAMA 2012), ARDS is defined as a type of acute diffuse lung injury associated with a predisposing risk factor, characterized by inflammation leading to increased pulmonary vascular permeability and alveolar collapse [20]. When emerging in response to sepsis, ARDS develops with an early inflammatory phase as well as extensive lung tissue necrosis. Next, myofibroblasts migrate to the wound and start proliferating as well as depositing collagen fibers, eventually culminating in progressive pulmonary fibrosis [21].

Therefore, the first event in the onset of ARDS is the impairment of the epithelial/endothelial barrier integrity. The integrity of the epithelial/endothelial barrier is fundamental in preventing alveolar edema formation and leukocyte extravasation (Figure 1). Key players in maintaining a balanced alveolar permeability are epithelial (E-cadherin) and vascular endothelial cadherin (VE-cadherin). Proinflammatory and permeability-inducing factors, particularly vascular endothelial growth factor (VEGF), tumor necrosis factor (TNF), or histamine, can induce Src-mediated phosphorylation of VE-cadherin at Tyr685, resulting in increased vascular permeability. Vascular endothelial growth factor (VEGF) stimulation mediates vascular permeability of alveolar endothelial cells, and pharmacological inhibition of PI3K counteracts it [22]. In addition, tumor necrosis factor receptor 2 (TNFR2), one of the 2 receptors for TNF-α, induces phosphorylation of IκBα kinase-β and subsequent activation of the canonical NF-KB pathway through PI3K/Akt activation [23]. Another signaling pathway linked to PI3K and important in the maintenance of the epithelial/endothelial barrier integrity is the renin/angiotensin pathway, which has emerged in the last years as a mechanism playing a role in the degree of lung injury, affecting vascular tone/permeability, epithelial cell survival, and fibroblast activation. In the renin/angiotensin pathway, the angiotensin I is eventually converted to angiotensin II (AngII), a molecule responsible for arterial vasoconstriction that can induce vascular damage at a high concentration. Interestingly, the lack of PI3Kγ reduces response to AngII in isolated vessels ex vivo and protects from AngII-induced vascular damage in vivo [24]. Therefore, the inhibition of this PI3K isoform is likely to be beneficial in the treatment of ARDS-induced vascular damage. Altogether, these findings suggest that PI3K inhibition may be considered as a potential vasculo-protective therapy in ARDS-induced impairment of epithelial/endothelial barrier integrity.

In ARDS, the second pathological event is a reduction in the alveolar fluid clearance (AFC). The epithelial layer inside the pulmonary alveoli is responsible for the accurate regulation of the alveolar fluid. The AFC is the active reabsorption process of sodium, chloride, and water from the air compartment to keep the lung dry. Several factors are known to be involved in reduced AFC, including a decreased density of Na/K ATPase on the basolateral membrane, interleukin-8 (IL-8) inhibition of β2-adrenergic receptor and TGF-β impairment of ENaC stability on the cell membrane surface. PI3K class IA may be involved in this process, as it regulates the trafficking of Na/K ATPase in the opossum renal epithelial cell line [25]. In addition, RAC-dependent PI3K signaling increases IL-8 production in lung epithelial cells [26].

The third and final event in ARDS is intra-alveolar fibrosis and diffuse alveolar damage (DAD). This is frequently seen in severe COVID-19 disease, where ARDS and lung fibrosis are common pathological consequences in many patients infected with severe SARS-CoV-2 [27]. Early analysis of COVID-19 patients discharged from the hospital suggests a high rate of lung function abnormalities linked to fibrosis. Upon discharge, 47% of patients were found with impaired gas transfer and 25% with reduced lung capacity, while the lung function was much worse in the severe form of the disease [28]. The fibrotic process is present in the two phases of the disease: during infection and after the patient has recovered [28]. Alveolar epithelium cell replacement by fibrotic tissue during and after the infection increases lung stiffening and induces the characteristic symptom of breath shortening in patients. Accordingly, pulmonary fibrosis is generally seen at autopsy in fatal cases of COVID-19 [29]. Among severe COVID-19 patients, ARDS is accompanied by uncontrolled fibroblast proliferation with PI3K/AKT/mTOR pathway activation. This pathway is a critical regulator of cell proliferation [13], hence its upregulation could result in the accumulation of fibroblasts and myofibroblasts, eventually leading to the excessive deposition of collagen and the progression of pulmonary fibrosis [30]. Accordingly, TGF-β signaling, a critical promoter of lung fibroblasts proliferation, differentiation and extracellular matrix production, can be modulated through PI3K inhibition [30]. In addition to the ubiquitously expressed PI3Kα and PI3Kβ, also PI3Kδ and PI3Kγ are functionally expressed in human lung fibroblasts and their selective inhibition suppresses fibroblast proliferation and differentiation. Thus, the downstream signaling of the TGF-β stimulation can be modulated through the inhibition of PI3Ks, specifically PI3Kγ and PI3Kα, eventually reducing ARDS-induced intra-alveolar fibrosis [31].

## 4. PI3K Inhibitors to Treat Lung Inflammation in Sepsis and ARDS

The class I PI3K represents an important pathway in several airway diseases associated to uncontrolled proliferation and immune dysregulation. Unfortunately, drugs designed to target PI3Ks turned out in clinical trials to be less tolerable than expected, due to systemic adverse effects [6,32]. The principal collateral effects of PI3K inhibitors are hyperglycemia, anorexia, nausea, vomiting, diarrhea, and stomatitis [33,34]. Strategies to avoid such complications are the development of isoform-specific PI3K inhibitors as well as the design of non-systemic drugs. The first approach aims at achieving maximal inhibitory doses while reducing the range of PI3K isoenzymes blocked, while the second minimizes systemic exposure and toxicity risks. Thus, cases of successful strategies that overcome systemic adverse effects exist and can be employed to treat lung inflammation. Table 1 shows examples of these molecules that could be potentially used to treat sepsis- and ARDS-induced lung inflammation.

Among the four PI3K isoforms, PI3Kδ already represents a valid target for treating various inflammatory airway disorders [35]. Accordingly, a central role in inflammatory processes is performed by PI3Kδ, which promotes the production and the release of pro-inflammatory cytokines, such as interleukins and interferon-γ [35]. Idelalisib, a PI3Kδ-selective inhibitor, together with ebastine, an antihistaminic, suppresses the release of pro-inflammatory cytokines (IL-1β, IL-8, IL-6 and TNF-α) by T-cells in airways and allergic diseases. Palma G. et al., suggest that idelalisib, alone or in combination with ebastine could be a promising therapeutic strategy for COVID-19-induced ARDS [35]. At the same time, another PI3K isoform, PI3Kγ, is crucially involved in chemoattractant-induced innate immune cells migration to the site of infection [34]. Duvelisib, an oral PI3Kδ- and γ-selective inhibitor, has been designed to block the contribution of the two isoenzymes and displays suitable properties for clinical development as an anti-inflammatory therapeutic. An ongoing phase II trial is examining the efficacy of duvelisib in hampering the immune system hyperactivation and, therefore, in reducing lung inflammation and ARDS in COVID-19 patients [36,37].

On the other hand, systemic treatment with PI3K inhibitors has turned out in clinical trials to cause severe on-target side effects including hyperglycemia and diarrhea [38]. Unfortunately, such side effects are also detected with isoform-specific inhibitors and even in drugs with selectivity for leukocyte-specific PI3K isoforms such as PI3Kδ- and γ. New therapeutic regimens/strategies must be designed to overcome such important unwanted effects.

One of these strategies, for example, is based on drug delivery restricted to the lungs and, therefore, avoiding exposure of critically sensitive tissues such as liver, heart and bowels. An inhibitor administered through inhalation is expected to have a better safety profile and may be appropriate for patients primarily affected by airway infections. Among the PI3K inhibitors, the most advanced in its clinical development is nemiralisib, an inhaled PI3Kδ-selective inhibitor that is under investigation in clinical trials as an immunomodulator to treat lung inflammation [39]. 

In line with this view, Campa et al., show that the inhaled prodrug PI3K inhibitor CL27c can reduce inflammation and improve lung function in models of asthma and irreversible pulmonary fibrosis. The reported pan-class I PI3K inhibitor reduces PI3K activation in lymphocytes, neutrophils, and bone-marrow-derived macrophages [40]. As a further safety feature, CL27c is a prodrug that is rapidly activated inside the cell, but this active compound is hydrophilic and is unable to cross the plasma membrane and diffuse outside target organs. In addition, if this active compound reaches the blood stream, it is captured by kidneys and secreted in the urine. Although not yet in clinical trials, further testing will soon define whether this drug can safely reach the patient.

Therefore, local delivery and improved safety profiles of PI3K inhibitors as immunomodulators may soon extend the therapeutic choices for sepsis- and ARDS-induced lung inflammation.

## 5. PI3K Inhibitors to Treat Sepsis- and ARDS-Induced Pulmonary Fibrosis

Sepsis- and ARDS-induced lung inflammation are frequently followed by the development of pulmonary fibrosis. The commonest pathological model of pulmonary fibrosis in mice is intra-tracheal bleomycin treatment, which mimics human ARDS and induces lung fibrosis by causing direct damage to the bronchoalveolar epithelium [43]. Mouse models of bleomycin-induced pulmonary fibrosis are characterized by increased PI3K signaling in the lungs and their inhibition improves pulmonary function and prevents fibrogenesis [44]. Accordingly, lung fibroblasts derived from patients affected by pulmonary fibrosis and treated with PI3K inhibitors show a reduction in the PI3K pathway measured through Akt phosphorylation and a consequent proliferation rate decrease in vitro [40]. Therefore, the PI3K/Akt pathway has gained growing interest in this field due to its key roles in lung fibroblast proliferation. In addition to the ubiquitously expressed PI3Kα and β, PI3Kδ and γ are functionally expressed in human lung fibroblasts. Remarkably, the results obtained with selective PI3K inhibitors demonstrated a major role of PI3Kγ and PI3Kα in TGF-β-induced fibroblast proliferation and differentiation [30]. Given the lack of effective therapies against sepsis- and ARDS-induced pulmonary fibrosis [18], targeting the TGF-β/PI3K axis represents a potential strategy to reduce lung fibroproliferation. Several therapeutic agents suppressing the PI3K/AKT pathway have been developed to treat pulmonary fibrosis but only some of them have entered clinical trials (Table 2). We next describe the development of PI3K inhibitors aimed at reducing lung fibroblast proliferation and report their strategy to decrease adverse effects. 

Hettiarachchi et al., describe a novel molecule consisting of a ligand to target fibroblast activation protein (FAP) conjugated to an inhibitor of PI3K. The treatment with the FAP-targeted PI3K inhibitor slows lung fibrosis, suppresses the production of hydroxyproline (major building block of collagen), reduces collagen deposition, and extends life in mouse models of pulmonary fibrosis [45].

A renowned inhibitor of the PI3K pathway, Omipalisib (GSK2126458), reduces primary human lung fibroblasts proliferation. A proof-of-mechanism trial is currently underway for this agent that represents a promising therapeutic agent to treat pulmonary fibrosis [46]. A phase I study is underway to assess the safety, tolerability, and pharmacokinetics of another class I PI3Ks and mTOR inhibitor, HEC68498. This inhibitor is highly selective and shows robust activity against fibrosis and inflammation (NCT03502902). However, to date, no relevant result has been obtained in the clinical study.

CL27c, a pan-class I PI3K inhibitor, can down-regulate the phosphorylation of AKT in mouse models of pulmonary fibrosis undergoing inhalation of the prodrug. Recent studies indicate that inhaled CL27c can protect from bleomycin-induced lung fibrosis by reducing inflammation and improving lung function. These results suggest that the local delivery of pan-PI3K inhibitor can effectively treat pulmonary fibrosis with the further advantage of reducing systemic on-target side effects.

## 6. Conclusions

Considering that the PI3K pathway represents a key node in the development of the fibrotic phenotype, inhibitors targeting these enzymes represent a promising pharmacological approach to the treatment of pulmonary fibrosis. However, the tolerability of such drugs is limited due to dose-limiting and on-target adverse effects; therefore, new strategies to address these drawbacks are urgently needed. New downstream targets in the pathway should be considered, especially if their inhibition can provide better safety profiles. Defining new targets controlling endothelial/epithelial barrier integrity, lung inflammation and TGF-α and TGF-β signaling in fibroblasts appears poised to provide future applications [54]. Considering that a major factor responsible for the mortality of sepsis and ARDS is the emergence of pulmonary fibrosis, the use of aerosolized anti-fibrotic compounds (Figure 1) limiting toxicity due to systemic exposure could also represent a game changer for effective and safe treatment.

Following infection, both sepsis and ARDS trigger lung inflammation often worsening in cytokine storm (top gray panel). The cytokine stimulation recruits leukocytes in the alveolar lumen, which is then chronically damaged and scarred. First, resident fibroblasts are gradually transformed in myofibroblasts. Next, in loco collagen deposition by myofibroblasts gives rise to lung stiffening-associated shortness of breath characteristic of pulmonary fibrosis. This pathological process is controlled by the TGF-β stimulation operating through the PI3K pathway, for which inhibitors are shown in the green and gray panels. 

## Figures and Tables

**Figure 1 biomedicines-10-00756-f001:**
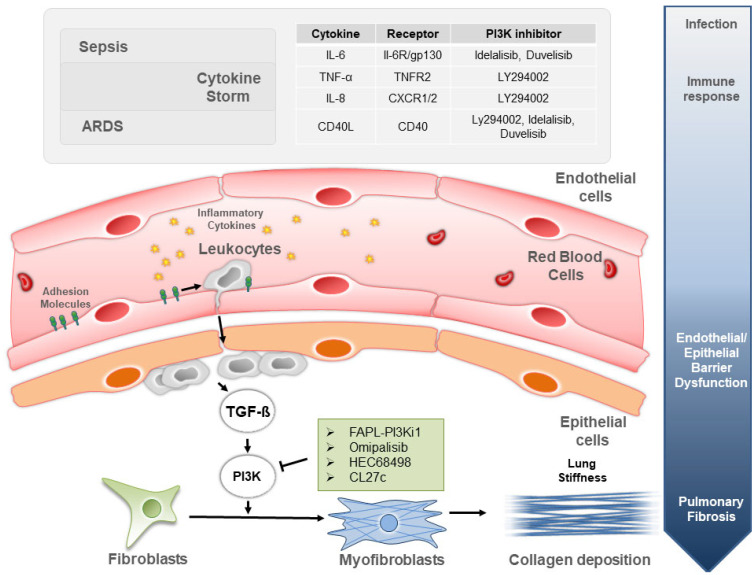
The PI3K signaling involvement in mechanisms and treatments of lung fibrosis induced by sepsis and ARDS.

**Table 1 biomedicines-10-00756-t001:** Selective and pan-class I PI3K inhibitors used as immunomodulators to suppress lung inflammation.

Agent	Mechanism/Target	Function	Common Adverse Events	Phase of Development	References
Idelalisib	PI3Kδ-selective inhibitor	Suppress the release of pro-inflammatory cytokines	Mild adverse events, unrelated to study medication	Phase I completed (Allergic Rhinitis) (NCT00836914)	[41]
Duvelisib	PI3Kδ- and γ-selective inhibitor	Inhibit innate and adaptive immune activity	Not described	Phase II active, not recruiting (COVID-19) (NCT04372602)	[42]
Nemiralisib	PI3Kδ-selective inhibitor	Reduces lymphoproliferation Anti-inflammatory	Any adverse event Post-inhalation cough	Phase II completed (APDS) (NCT02593539) Phase II terminated (COPD) (NCT03345407)	[39]
CL27c	Pan-class I PI3K inhibitor	Anti-inflammatory and anti-fibrotic	Not described	Pre-clinical	[40]

PI3K: phosphatidylinositol 3-kinase; PI3Kδ: phosphatidylinositol 3-kinase δ; PI3Kγ: phosphatidylinositol 3-kinase γ; COVID-19: coronavirus disease 2019; APDS: activated PI3Kδ syndrome; COPD: chronic obstructive pulmonary disease.

**Table 2 biomedicines-10-00756-t002:** PI3K pathway inhibitors of pulmonary fibrosis.

Agent	Mechanism/Target	Function Description	Common Adverse Events	Phase of Development	References
GSK2126458	PI3K/mTOR inhibitor	Anti-fibrotic	Diarrhoea, hyperglycaemia, nausea	Phase I completed (NCT01725139)	[47]
HEC68498	PI3K inhibitor	Anti-fibrotic and anti-inflammatory	Not described	Phase I completed (NCT03502902)	\
FAPL-PI3Ki1	PI3K/mTOR inhibitor	Anti-fibrotic	Not described	Pre-clinical	[45]
CL27c	Pan-class I PI3K inhibitor	Anti-inflammatory and anti-fibrotic	Not described	Pre-clinical	[40]
PX-866	Pan-PI3K inhibitor	Anti-fibrotic	Rash, hyperglycemia, transaminase elevations	Pre-clinical	[48]
LY294002	Akt inhibitor	Anti-fibrotic	Not described	Pre-clinical	[49]
ASV	TβR1/PI3K/AKT pathway inhibition	Anti-fibrotic	Raised total bilirubin and rash	Pre-clinical	[50]
Hyperoside	AKT/GSK3β inhibitor	Anti-fibrosis	Not described	Pre-clinical	[51]
Ligustrazine	PI3K/AKT/mTOR pathway inhibition	Anti-fibrotic	Edema, hypertension, gastrointestinal bleeding	Pre-clinical	[52]
Derivatives of 4-methylquinazoline	PI3K inhibitor	Anti-fibrotic and anti-inflammatory	Not described	Pre-clinical	[53]

PI3K: phosphatidylinositol 3-kinase; mTOR: mammalian target of rapamycin; FAPL-PI3Ki1: fibroblast activation protein targeted PI3K inhibitor; TβRI: type I subunit of the TGF-β receptor, also known as ALK5; ASV: astragaloside IV; GSK3β: glycogen synthase kinase 3β.

## Data Availability

The study did not report any data.

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
