# Peer review of "PI3K Signaling in Mechanisms and Treatments of Pulmonary Fibrosis Following Sepsis and Acute Lung Injury"

_biomedicines, 2022, doi:10.3390/biomedicines10040756_

Round 1
Reviewer 1 Report
The study of the mechanisms of pulmonary fibrosis that is induced, among other things, by a new coronavirus infection is an extremely timely and important at the present time. The authors presented a potentially interesting review about the role of the PI3K/Akt pathway in the development of pulmonary fibrosis and summarized some data on PI3K inhibitors targeting this disease. Although this review is informative and useful for readers, some changes need to be made before it is published.
Major points
The review should be structured by highlighting the PI3K/Akt pathway.
The authors paid more attention to the pathogenesis of fibrotic processes, but the mechanisms of PI3K/Akt involvement in this process not clearly presented. The authors can revise the sections of the review and include, for example, the sections on the PI3K signaling involvement in infections and the immune response. Reference to Figure 1 should be included in these sections. In addition, the authors can make a separate section on PI3K inhibitors targeting fibrotic processes.
Minor points
- In the Introduction, the authors could provide some references (for example, line 25 line; 47; line 58).
- The acronym «PI3K» should be specified in full already at the introduction (in the line 51).
- Line 299. The acronym «PI3K» to be removed.
- Line 57 and line 92. Add acronyms «TGF-beta» and «TNF-alfa», respectively.
- The authors should carefully check all review acronym and remove any repeats.
Reviewer 2 Report
In this review, the authors have nicely summarized the role of PI3K/AKT signaling in fibrotic processes and potential inhibitors for mitigating fibrotic progression.
The section “PI3K inhibitors in regulating the immune response” would be better with a table summary including the underlining mechanisms and potential disadvantages.
Reviewer 3 Report
The authors provided a review article to discuss the role of PI3K signaling in mechanisms and treatments of pulmonary fibrosis following sepsis and acute lung injury. Although the PI3K signaling play an important role on the pathogenesis of pulmonary fibrosis, the causes of pulmonary fibrosis following sepsis and acute lung injury are multifactorial. For example, inflammation and ventilator-induced lung injury may contribute to pulmonary fibrosis during ARDS progression.
Some sections of the main text were not correlated well with the title of the manuscript (i.e., pulmonary fibrosis following sepsis and acute lung injury), for example, cytokine storm, COVID-19, and idiopathic pulmonary fibrosis (IPF). Not all sepsis and ARDS patients would have cytokine storm, which may also not be an essential factor contributing to pulmonary fibrosis. IPF is a chronic, progressive fibrotic interstitial pneumonia, and the mechanisms and treatments of pulmonary fibrosis following sepsis and ARDS may not be the same as IPF. The antifibrotic agents for IPF are not currently proved to be an evidenced-base therapy for COVID-19 patients.
The structure of manuscript is not well organized and not comprehensive. Some sentences were hard to be interpreted. Finally, I had some comments.
Major comments:
- The potential mechanism driving fibroproliferation in sepsis and ARDS should be described more clearly except PI3K signaling or cytokine storm.
- The content of Table 1 should be added, like function description (ex: anti-fibrosis, anti-inflammation…), clinical trial identifier (NCT number), common adverse events, and reference.
- Some grammatical mistakes were noted in the manuscript, and suggested a medical editor who is a native English speaker to edit this manuscript.
- Page 3, line 124: AECC definition should be revised to Berlin definition (JAMA 2012)
- Page 6, line 272-286: suggested to be deleted, not correlated with the topic of this manuscript.
- Page 8, the figure legend (line 338-350) should be shortened and summarized.
Minor comments:
- The reference numbers in the manuscript should be placed before the punctuation. For example [1], [1–3] or [1,3].
- An abbreviation should be used at the first occurrence of the sentence in the text, like ARDS, SARS-CoV-2, PI3K.
- Page 1, line 32, 42; page 3, line 112: acute respiratory “disease” syndrome, disease should be revised to distress.
- Page 6, line 305: What is ” top38lerability” meant?
- The full term of abbreviations should be added in the footer of Table and Figure legend.
- The style of the references should be adjusted as MDPI template.
Reviewer 4 Report
- I don't see any references in the Introduction section.
- In the abstract abbreviation, ARDS is not defined first.
- The authors have given subtle information in table 3, but it would be valuable to add information about the development of these inhibitors.
- It would be great to add some discussion points about the additional
considerations for corticosteroids role on cytokine storm and inflammation. - The author should also discuss what is the limitation of the current treatments and what should be future directions for this field.
- Authors should include a section with the discussions of the molecular
mechanisms underlying kinase-inhibitor induced cardiotoxicity or for these adverse events, whenever possible. Many studies have reported the mechanism behind the cardiotoxicity of these inhibitors.
Round 2
Reviewer 1 Report
The authors have addressed all reviewer comments and revised the manuscript in detail.
However, the authors should remove the reference to Figure 1 in conclusion section and remove also the previous version of Figure 1.
The manuscript can be accepted for publication after this correction.
Reviewer 3 Report
I read the revised manuscript. The authors had clealy responsed to my previous suggestions. I had no further comments.